# UNVEILING CONTEXT-AWARE CRITERIA IN SELF-ASSESSING LLMS

## ABSTRACT

The use of large language models (LLMs) as evaluators has garnered significant attention due to their potential to rival human-level evaluations in long-form response assessments. However, current LLM evaluators rely heavily on static, human-defined criteria, limiting their ability to generalize across diverse generative tasks and incorporate context-specific knowledge. In this paper, we propose a novel Self-Assessing LLM framework that integrates Context-Aware Criteria (SALC) with dynamic knowledge tailored to each evaluation instance. This instance-level knowledge enhances the LLM evaluator's performance by providing relevant, context-aware insights that pinpoint the important criteria specific to the current instance. Additionally, the proposed framework adapts seamlessly to various tasks without relying on predefined human criteria, offering a more flexible evaluation approach. Empirical evaluations demonstrate that our approach significantly outperforms existing baseline evaluation frameworks, yielding improvements ranging from 5% across a wide variety of datasets. Furthermore, by leveraging knowledge distillation techniques, we fine-tuned smaller language models for criteria generation and evaluation, achieving comparable or superior performance to larger models with much lower cost. Our method also exhibits a 5% improvement on the Alpaca leaderboard when employed for preference data generation in Direct Preference Optimization (DPO), underscoring its efficacy as a robust and scalable evaluation framework.

## 1 INTRODUCTION

Evaluating the quality of machine-generated text has long been a significant challenge in Natural Language Processing (NLP), particularly with the advent of Large Language Models (LLMs), where a comprehensive understanding of their capabilities and functions is crucial. Traditional metrics such as BLEU (Papineni et al., 2002) and ROUGE (Lin, 2004) for lexical analysis, and BERTScore (Zhang et al., 2019) and BARTScore (Yuan et al., 2021) for semantic evaluation, often fall short of capturing the full depth and nuance of human judgment. Recently, LLM-based evaluator have gained significant attention due to its alignment with human judgements for evaluation, utilizing zero-shot or few-shot instructions (Chiang et al., 2023; Dubois et al., 2024; Liu et al., 2023).

Despite advancements in LLM evaluators, which can effectively follow human-prepared criteria to assess generative tasks aligned with human preferences, they often rely heavily on static, pre-defined criteria that are applied uniformly across all evaluation instances. This approach struggles to generalize across diverse tasks (Liang et al., 2022; Lee et al., 2022; Krishna et al., 2023; Min et al., 2023) and fails to provide the context-aware knowledge crucial for specific evaluation instances. However, we observe that context-aware criteria are essential for providing effective guidance across diverse evaluation instances. For instance in the example defined in figure 1, while the initial response mentions climate change and melting ice, it lacks detail about how these factors specifically affect polar bear populations. Pre-defined criteria might only check for surface-level accuracy, missing out on the need for a deeper explanation of hunting challenges, energy expenditure, and survival rates, as seen in the reference.

In response to this challenge, we propose Self-Assessing LLMs with Autonomous Criteria Generation (SALC), a framework designed to allow LLMs to autonomously generate the criteria for evaluating the responses for the given instruction. SALC shifts the paradigm from human-centric

---

**Instruction:** Summarize the impact of climate change on polar bear populations in a few sentences

**Reference:**
Climate change is leading to the rapid melting of sea ice, which polar bears rely on for hunting seals, their primary food source. The loss of this critical habitat is causing polar bears to travel longer distances and expend more energy, leading to decreased survival rates and reproductive success.

**Response:**
Climate change is affecting polar bears. Their ice habitats are melting, making it harder for them to find food. This could lead to a decline in their population.

**Self-Assessing Criteria**

1.) **Relevance to Instruction** - Does the response focus on the specific instruction of summarizing the impact of climate change on polar bear populations? Does it provide relevant information about the effects of climate change?

2.) **Completeness** - Does the response sufficiently cover key aspects of the topic? Are any critical details or consequences of climate change on polar bears (e.g., loss of habitat, challenges in hunting) missing?

3.) **Clarity and Coherence** - Is the response easy to understand? Is it presented in a clear and logical manner that flows naturally and provides a coherent explanation of the topic?

4.) **Conciseness and Brevity** - Given the instruction to summarize in "a few sentences," does the response maintain brevity without sacrificing important information? Is the length appropriate for the instruction?

5.) **Accuracy** - Is the information in the response factually accurate based on the reference? Does it correctly describe the impact of climate change on polar bears, such as their dependence on ice habitats and food scarcity?

Figure 1: Illustrative Example of SALC's Criteria Generation Process

evaluation to a model-driven approach, leveraging the language understanding and reasoning abilities of advanced LLMs, such as GPT-4. By generating task and context specific evaluation criteria in real time, SALC enables more accurate and contextually appropriate assessments of LLM outputs. SALC operates in two distinct settings: the absolute setting – where the model generates criteria based on an instruction, a reference answer, and a response; and the relative setting – where the model compares multiple responses to the same instruction. This dual configuration allows SALC to evaluate a broader range of tasks and responses, adapting its criteria based on the complexity and nature of the specific instruction. This flexibility overcomes the limitations of traditional evaluation methods, which rely on manually defined static criteria that may not align with the nuances of each task.

In the given example in figure 1, dynamic criteria would provide a more comprehensive and accurate evaluation by adjusting to the specific context and depth required for a high-quality response. (1) *Surface-Level Accuracy* : The initial response is correct but lacks detail. It mentions "climate change" and "melting ice" but doesn't elaborate on their consequences. Static criteria might only check for these basic concepts, missing out on the need for further explanation. (2) *Depth and Specificity*: The reference response offers more depth, explaining how melting ice forces polar bears to travel longer distances, affecting their survival and reproduction. Dynamic criteria would recognize this added value, rewarding responses that offer specific, contextually relevant details. (3) *Context Awareness*: Dynamic criteria can adjust to the complexity of the subject. In this case, a more nuanced explanation of the ecological impacts, like habitat loss reducing hunting efficiency and increasing energy expenditure, would be rewarded, whereas static criteria might not account for these subtleties. (4) *Clarity and Comprehensiveness*: Beyond factual accuracy, dynamic criteria would evaluate how well the response communicates the issue. The reference response clearly connects habitat loss to biological impacts, providing a more complete picture. Dynamic criteria ensure responses are evaluated not just for correctness but also for their clarity and thoroughness. (5) *Flexibility* : Dynamic criteria are adaptable to different levels of detail based on the audience. A concise, simpler response might be appropriate for a general audience, while an in-depth explanation is better suited for expert evaluations. This flexibility allows for fair assessments based on context, ensuring that the criteria align with the purpose of the evaluation.

Thus, dynamic criteria ensure that responses are evaluated not just for surface-level accuracy but for how well they explain, contextualize, and convey complex issues, leading to a more thorough and fair evaluation process.

Furthermore, we introduce SALC-Tune, which fine-tunes smaller models for both criteria generation and evaluation, addressing the challenges posed by the proprietary nature of LLMs. Both the evaluation criteria and feedback are autonomously generated using the SALC framework on the Feedback Collection Dataset (Kim et al., 2023), with GPT-4 serving as the underlying model. The dataset generated by SALC is then used to fine-tune two models: FT-Criteria for criteria generation and FT-Judge for evaluation. By distilling knowledge from GPT-4, these smaller models capture the quality of the criteria and feedback generated by the larger model, delivering competitive performance while being more efficient in size. This underscores the effectiveness of our SALC-based fine-tuning approach for both evaluation and feedback generation.

Additionally, SALC demonstrates its effectiveness when applied as a reward model for preference-data generation in Direct Preference Optimization (DPO). By autonomously generating evaluation criteria and providing more accurate, instruction-relevant assessments of responses, SALC boosts performance in DPO by at least 5% over existing baselines, showcasing its potential not only in standard LLM evaluations but also in enhancing preference-based learning tasks.

To that end, our Key Contributions are as follows:

- **Adaptive Evaluation Criteria Generation**: SALC introduces a framework where LLMs autonomously generate instruction-specific evaluation criteria, enabling more adaptive and accurate assessments compared to static, predefined human-generated standards.

- **Efficient Evaluation with Cost-effective Fine-tuned Models**: By fine-tuning smaller models with GPT-4-generated criteria, feedback and score generated based on the generated criteria, SALC demonstrates that even compact models can outperform larger ones, offering an open source, efficient and scalable evaluation solution.

- **Improved DPO Performance**: SALC enhances preference-data-generation in DPO tasks, delivering significant improvements over traditional methods and setting new baselines for preference optimization using autonomous evaluation criteria.

These contributions establish SALC as a robust, flexible, and efficient framework for LLM evaluation, reducing reliance on human biases and enabling more accurate and task-aware assessments.

## 2 RELATED WORKS

The evaluation of Large Language Models (LLMs) has evolved from traditional metrics like BLEU Papineni et al. (2002) and ROUGE Lin (2004), which primarily focus on surface-level lexical similarity, to more sophisticated semantic-based approaches. These newer methods, such as BLEURT Sellam et al. (2020), BERTScore Zhang et al. (2019), and BARTScore Yuan et al. (2021), evaluate outputs at a deeper level, capturing more meaningful aspects of text. However, these model-based metrics, while significant improvements, remain static and limited by their dependence on fixed reference criteria, which makes them less adaptable to diverse, task-specific requirements.

Recent efforts have explored LLMs themselves as evaluators, particularly in preference-based scenarios. Models like Alpaca-Farm Du et al. (2023) take this further by allowing the model to choose better responses through its own judgments, marking a shift toward model-driven evaluations. Additionally, open-source LLMs like PROMETHEUS Kim et al. (2023; 2024) have emerged to offer customizable, fine-grained evaluation capabilities on par with proprietary models like GPT-4. PROMETHEUS leverages its own dataset of score rubrics, instructions, and responses, achieving strong performance in correlation with human judgments and surpassing models like ChatGPT in specific tasks. This development offers a scalable and cost-effective solution for practitioners needing custom evaluation criteria, particularly when dealing with large-scale evaluation tasks.

Further advancements in fine-grained evaluation are exemplified by FLASK Ye et al. (2023b), which decomposes coarse-level scoring into more granular skill-based evaluations. This protocol improves interpretability by focusing on instance-specific skill requirements in instructions, enhancing both model- and human-based evaluations. Research shows that fine-grained evaluation, as provided by FLASK, offers a more comprehensive view of model performance and increases the reliability of evaluations across multiple tasks and datasets.

Our proposed framework, SALC, advances LLM evaluation by autonomously generating task-specific criteria, unlike static, human-defined rubrics. SALC dynamically adapts to both absolute and relative evaluation contexts, mitigating bias and improving correlation with human judgments. Existing approaches like LLM as Judge (Zheng et al., 2023) use fixed evaluation criteria, which fail to capture nuances across tasks. In contrast, Flask defines 12 skill sets with static, human-curated rubrics, limiting adaptability. Flask selects top skills for evaluation but relies on predefined prompts, making it less flexible. Similarly, Prometheus also requires human input for every new evaluation scenario. SALC overcomes these limitations by enabling criteria generation specific to any task or context without relying on predefined, rigid rubrics.

## 3 METHODOLOGY

The SALC (Self-Assessing LLMs with Autonomous Criteria Generation) framework operates through several key stages, designed to enhance the evaluation of LLM outputs by autonomously generating task-specific criteria. The pipeline described in figure 2 is structured as follows: (i) *Criteria Generation Stage* : SALC begins with the autonomous generation of evaluation criteria tailored to the specific instruction. Unlike traditional methods that rely on predefined rubrics, SALC, dynamically generates criteria based on the instruction context. This allows the framework to capture the nuances of different responses across diverse domains. (ii) *Evaluation Stage*: Once the criteria are generated, SALC evaluates model outputs in two settings: Absolute Evaluation and Relative Evaluation. For absolute evaluation, each response is evaluated independently against the generated criteria, scoring it based on task-specific metrics. For relative evaluation, SALC compares responses to each other, ranking them according to their alignment with the generated criteria. This relative scoring reduces the noise that often arises from absolute scoring alone. (iii) *Fine-Tuning Open-Source Models*: SALC incorporates fine-tuning of smaller, open-source models for both criteria generation and evaluation. FT-Criteria handles criteria generation, while FT-Judge is fine-tuned to evaluate responses. This fine-tuning is conducted using knowledge distillation from GPT-4 into models like llama-7b and llama-13b, enabling efficient yet effective performance. (iv) *Preference Data Generation*: In addition to direct evaluation, SALC is applied to generate preference data for Direct Preference Optimization (DPO). The use of dynamic context-aware generated criteria improves the quality of preference data, leading to enhanced fine-tuning outcomes in LLM performance.

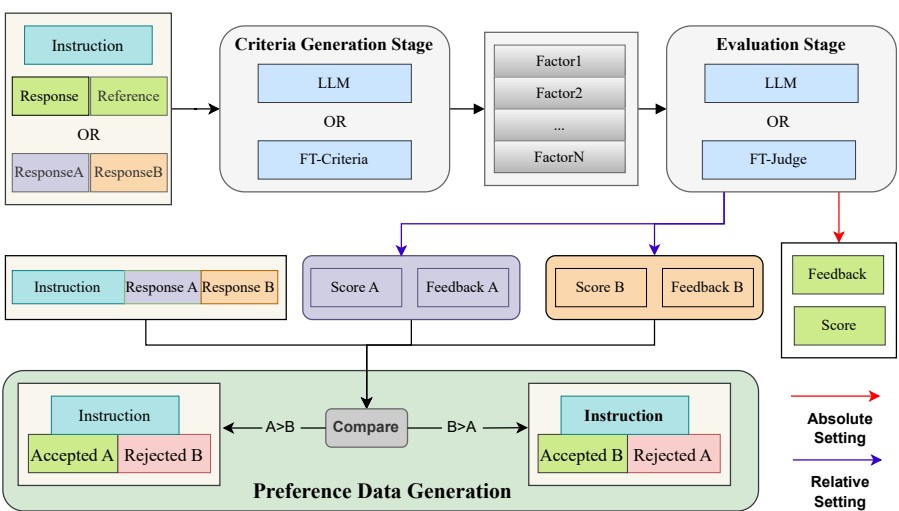

Figure 2: Overview of the SALC Pipeline

### 3.1 CRITERIA GENERATION

The first step in our methodology involves using an LLM, denoted as $M$, to autonomously generate evaluation criteria in a zero-shot setting. Given a prompt p, which consists of three core elements – (1) a user instruction $\alpha$, (2) a reference answer $r$ (in the absolute setting), and (3) the response to

be evaluated $o$ (or multiple responses $r_a$ and $r_b$ in the relative setting) – the LLM generates a set of factors which are used as the evaluation criteria $C$. This process occurs without predefined human input or additional training, allowing the LLM to create a criteria framework based solely on its understanding of the context. In absolute grading, we use the following inputs:

- *User Instruction*: The user instruction given to the model, denoted by $\alpha \in \mathcal{I}$ where $\mathcal{I}$ represents set of Instructions (e.g., "Summarize the following text").
- *Reference Answer*: The expected or ideal response to the instruction, denoted by $r \in \mathcal{R}$ where $\mathcal{R}$ represents set of references (e.g., a human-written summary).
- *Response to be Evaluated*: The actual output produced by the LLM for evaluation, denoted by $o \in \mathcal{O}$ where $\mathcal{O}$ represents set of responses.

Whereas, in relative grading setting, we leverage the following as inputs:

- *User Instruction*: The task prompt given to the model, denoted by $\alpha \in \mathcal{I}$ (e.g., "Summarize the following text").
- *Response A*: First response provided for comparative evaluation, denoted by $o_a \in \mathcal{O}$ where $\mathcal{O}$ represents set of responses..
- *Response B*: Second Response provided for comparative evaluation, denoted by $o_b \in \mathcal{O}$ where $\mathcal{O}$ represents set of responses..

The model $M$ generates a set of evaluation criteria $C = \{c_1, c_2, \ldots, c_n\}$, where $c_j$ represents a factor (such as fluency, relevance, coherence, etc.) based on which the response $o$ is evaluated. The criteria $C$ are thus generated based on the instruction $\alpha$, the reference answer $r$, and the model response $o$ (or multiple responses $r_a$ and $r_b$ in the relative setting), without any predefined human standards, which can be formalized as:

$$C = f_M(\alpha, r, o) \ \text{ or } \ C = f_M(\alpha, o_a, o_b) \tag{1}$$

Here, $f$ represents the internal reasoning and language understanding function of the LLM. Key reasons why LLM-generated criteria are effective:

- Contextual Adaptation: The LLM dynamically adjusts its criteria based on the task, ensuring relevance, much like a human would adapt depending on context.
- Multi-Dimensional Judgment: $f$ evaluates factors such as correctness, style, and logical flow simultaneously, offering holistic assessments similar to human evaluators.
- RLHF Alignment: Fine-tuning with RLHF equips the model with a deeper understanding of human preferences, allowing it to generate nuanced criteria.

By using $f$, SALC reduces the reliance on predefined metrics, offering flexible, adaptive evaluation on par with human judgment.

### 3.2 ASSESSING RESPONSES

After generating the evaluation criteria $C$, the LLM $M$ uses these factor $c_j$ to assess the quality of the response $o$ given the instruction $\alpha$ and reference answer $r$. The evaluation is performed in a multi-step process, where each factor $C_j$ is applied to the response $o$. The LLM $M$ provides feedback $f_j$ on each criterion $c_j$, assigning a final score $S \in [1, 5]$ to the response based on how well it satisfies criteria $C$.

Overall process can be explained as this: For Absolute Setting let $s_j(\alpha, c_j, r, o)$ represents the score for response $o$ under the criterion $c_j$, where $s_j \in [1, 5]$ represents integer score. The overall score $S$ for the response $o$ is the aggregate score across all the criteria:

$$S(\alpha, C, r, o) = \sum_{j=1}^{n} \beta_j s_j(\alpha, c_j, r, o) \tag{2}$$

where $\beta_j$ are weights associated with each criterion, and $\sum_{j=1}^{n} \beta_j = 1$. These weights are based on the importance of the criteria for the given task given by the LLM internally.

For Relative Setting let $s_{aj}(\alpha, c_j, r_a)$ represents the score for response $o_a$ under the criterion $c_j$ and $s_{bj}(\alpha, c_j, r_b)$ represents the score for response $o_b$ under the criterion $c_j$, where $s_{aj} \in [1, 5]$ and $s_{bj} \in [1, 5]$ represents integer score. The overall score $S_a$ and $S_b$ for the responses $o_a$ and $o_b$ respectively is the aggregate score across all the criteria:

$$S_a(\alpha, C, o_a) = \sum_{j=1}^{n} \beta_{aj} s_{aj}(\alpha, c_j, o_a) \tag{3}$$

$$S_b(\alpha, C, o_b) = \sum_{j=1}^{n} \beta_{bj} s_{bj}(\alpha, c_j, o_b) \tag{4}$$

where $\beta_{aj}$ and $\beta_{bj}$ are weights associated with each criterion for responses $o_a$ and $o_b$ respectively, and $\sum_{j=1}^{n} \beta_{aj} = 1$ as well as $\sum_{j=1}^{n} \beta_{bj} = 1$. These weights are based on the importance of the criteria for the given task given by the LLM internally.

Above process is adaptive and context-aware, allowing the model to provide a more nuanced and meaningful assessment of LLM-generated content.

### 3.3 FINE-TUNING USING THE FEEDBACK COLLECTION DATASET

For the fine-tuning process, we utilized the Feedback Collection Dataset $\mathcal{D}$, which was originally used in training the Prometheus Model Kim et al. (2023). However, we removed the score rubric, score and feedback provided in this dataset and instead generated our own evaluation criteria using GPT-4 using our method SALC. These newly generated factors were then employed to evaluate the quality of responses and provide corresponding feedback.

#### 3.3.1 CRITERIA AND FEEDBACK GENERATION

Using $\mathcal{L}$, we generated a criteria $C$ for each instruction $\alpha$, reference $r$, and response $o$ within the dataset $D$. These factors were then used to assess the quality of the responses and generate detailed feedback $F$, including final scores $S$ based on the factors generated.

$$\mathcal{L}(\alpha, r, o) \rightarrow C \tag{5}$$
$$\mathcal{L}(\alpha, C, r, o) \rightarrow F, S \tag{6}$$

#### 3.3.2 FINE-TUNING MODELS ON GENERATED FACTORS AND FEEDBACK

We utilized the criteria $C$ generated by GPT-4 to fine-tune small language model, enabling it to autonomously generate evaluation factors similar to those produced by GPT-4. Additionally, we fine-tuned another small language model on the feedback generated by GPT-4, allowing it to produce feedback $F$ and scores $S$ that closely align with GPT-4's assessments. This fine-tuning process enhances the model's ability to evaluate responses and generate feedback with a level of quality similar to GPT-4, enabling autonomous and high-quality evaluation of LLM outputs. This approach also facilitates open-sourcing the models, as we are no longer reliant on GPT-4's closed nature. Moreover, despite a negligible drop in performance compared to GPT-4, our fine-tuned models—using only 13B parameters—outperform many state-of-the-art (SOTA) open-source models, which typically operate with much larger architectures (e.g., 175B parameters). This efficiency gain makes our solution highly competitive in terms of both performance and scalability.

## 4 EXPERIMENTAL RESULTS

We evaluated SALC's performance against other evaluation strategies across various tasks, including absolute and relative grading setting and its use as a reward model for RLFH. Our experiments

utilized datasets such as Vicuna Bench (Chiang et al., 2023) , MT-Bench (Zheng et al., 2023) , Flask Eval (Ye et al., 2023b) , Alpaca Eval (Dubois et al., 2024) , HHH Alignment (Askell et al., 2021) , and Feedback Collection (Ye et al., 2023b) . We compared SALC with baseline methods including LLM as a judge (Zheng et al., 2023), Prometheus (Kim et al., 2024), and various pre-trained LLMs and SLMs. Evaluation metrics included accuracy (for ranking scenarios), correlation (for absolute grading), similarity (for criteria generation), and alignment (for HHH dataset). We used metrics such as Accuracy, F1-score, Pearson, Kendall-Tau, and Spearman correlations, as well as Rouge, Bleu, and BERT for textual similarity. The experiments aimed to comprehensively assess SALC's effectiveness in generating adaptive criteria and improving evaluation capabilities of pre-trained language models. Additional details on dataset, evaluation strategy and experimental setup is mentioned in Appendix B.

## 4.1 SALC PERFORMANCE ON PRETRAINED LLMS

We compared the performance of SALC against LLM as judge Ye et al. (2023a) and Prometheus methods using 3 widely used LLM models (GPT-3.5, GPT-4 and GPT-4o). To demonstrate the generalizability of SALC, we conduct experiments on three benchmark evaluation datasets: HHH Alignment, Alpaca Eval and MT Bench. In in Table 1, we demonstrate the Accuracy and F1-score between the ground truth on preferred responses and the respective evaluator's preference. SALC consistently outperformed the baseline methods across all the settings demonstrating its potential to enhance the reliability and safety of LLM evaluation. With GPT-3.5 acts as a downstreaming model, SALC improves the F1-score and Accuracy by at least 5% and 2.5% over both the LLM as judge and Prometheus baselines on HHH Alignment dataset. These significant gains highlight SALC's effectiveness in improving model alignment, particularly in sensitive domains requiring safety, honesty and helpfulness guarantees. With GPT-4, SALC consistently improvements the performance across all the benchmark datasets. On the HHH Alignment task, SALC improves the F1-score and Accuracy by at least 1.8% and 1.5%, respectively. Given the superior performance of pre-trained GPT-4 in reasoning capability, these improvements are significant enhancement in the model's ability to align with human values and expectations. With GPT-4o being larger and more powerful pre-trained model, we observe the most substantial benefits for SALC. On the HHH Alignment benchmark, SALC demonstrated a remarkable 5.4% improvement in both F1-score and accuracy compared to the LLM as judge, and a 2.9% improvement in F1-score over Prometheus. For the Alpaca Eval and MT Bench benchmarks, SALC provides consistent 2-3% increases in both F1-score and accuracy over baseline methods. These significant performance gains underscore SALC's ability to leverage the increased capacity of larger models. Additional experimental results on the superior performance of SALC on absolute grading scenario are mentioned in Appendix C.1.

| Evaluator LM | HHH Alignment | | Alpaca Eval | | MT Bench | |
|---|---|---|---|---|---|---|
| | F1-Score | Accuracy | F1-Score | Accuracy | F1-Score | Accuracy |
| GPT-35 (LLM as judge) | 0.777 | 0.776 | 0.462 | 0.543 | 0.4494 | 0.5504 |
| GPT-35 (Prometheus) | 0.784 | 0.792 | 0.509 | 0.511 | 0.465 | 0.534 |
| GPT-35 (SALC) | 0.821 | 0.811 | 0.521 | 0.538 | 0.5215 | 0.5564 |
| GPT-4 (LLM as judge) | 0.890 | 0.884 | 0.5697 | 0.5635 | 0.526 | 0.633 |
| GPT-4 (Prometheus) | 0.883 | 0.887 | 0.545 | 0.535 | 0.521 | 0.621 |
| GPT-4 (SALC) | 0.906 | 0.899 | 0.5752 | 0.5701 | 0.543 | 0.633 |
| GPT-4o (LLM as judge) | 0.890 | 0.885 | 0.557 | 0.562 | 0.510 | 0.632 |
| GPT-4o (Prometheus) | 0.912 | 0.914 | 0.550 | 0.552 | 0.534 | 0.627 |
| GPT-4o (SALC) | 0.938 | 0.933 | 0.584 | 0.577 | 0.5643 | 0.6463 |

Table 1: Agreement of Evaluator Language Models with different baselines across different benchmarks

## 4.2 SALC-TUNE ON FEEDBACK COLLECTION TEST SET

To understand the performance of SALC on cost-effective small LMs, we have fine-tuned open-sourced models and demonstrate their performance on Feedback collection test dataset for two tasks: (a) automated criteria generation, and (b) automated response evaluation.

**Criteria Generation Results** For understanding the quality of automatically generated criteria against the reference criteria generated by GPT-4, we used standard textual similarity metrics such as

Rouge (R1, R2, RL), Bleu, and Bert Score. As shown in Table 2, our finetuned models, FT-Criteria-7b and FT-Criteria-13b, significantly outperform pretrained baseline models (including llama-7b-chat, llama-13b-chat, and GPT-3.5-turbo) across both in-domain and out-of-domain test sets.

| Criteria LM | In-Domain Test | | | | | Out-of-Domain Test | | | | |
|---|---|---|---|---|---|---|---|---|---|---|
| | Rouge | | | Bleu | Bert | | Rouge | | Bleu | Bert |
| | R1 | R2 | RL | | | R1 | R2 | RL | | |
| LLaMA2-7b-chat | 0.445 | 0.178 | 0.259 | 0.111 | 0.859 | 0.455 | 0.186 | 0.257 | 0.116 | 0.861 |
| LLaMA2-13b-chat | 0.426 | 0.175 | 0.243 | 0.118 | 0.865 | 0.431 | 0.179 | 0.246 | 0.121 | 0.868 |
| LLaMA2-70b-chat | 0.426 | 0.175 | 0.243 | 0.118 | 0.865 | 0.431 | 0.179 | 0.246 | 0.121 | 0.868 |
| FT-Criteria-7b | 0.603 | 0.384 | 0.406 | 0.313 | 0.915 | 0.604 | 0.384 | 0.408 | 0.313 | 0.915 |
| FT-Criteria-13b | 0.624 | 0.405 | 0.429 | 0.337 | 0.921 | 0.627 | 0.405 | 0.43 | 0.340 | 0.922 |
| GPT-35-turbo | 0.534 | 0.238 | 0.304 | 0.270 | 0.896 | 0.536 | 0.261 | 0.322 | 0.273 | 0.900 |
| GPT-4-turbo | 0.582 | 0.277 | 0.326 | 0.321 | 0.906 | 0.589 | 0.282 | 0.328 | 0.326 | 0.907 |
| GPT-4o | 0.590 | 0.286 | 0.332 | 0.330 | 0.908 | 0.596 | 0.291 | 0.335 | 0.336 | 0.909 |

Table 2: Comparison of Criteria Generation Models on Feedback Collection Test Set

**Evaluation Results**    For evaluating generated feedback, we compute the standard correlation metrics (e.g., Pearson, Kendall-Tau, and Spearman) between our models' evaluation score and the reference GPT-4 score. As illustrated in Table 3, our finetuned evaluators, FT-Judge-7b and FT-Judge-13b, show remarkable correlation with GPT-4 as the reference for both in-domain and out-of-domain test sets, while outperforming even large models such as GPT-4o and GPT-4-Turbo.

| Evaluator LM | Feedback Collection Test set | | | | | |
|---|---|---|---|---|---|---|
| | In-Domain Test | | | Out-of-Domain-Test | | |
| | Pearson | Kendall-Tau | Spearman | Pearson | Kendall-Tau | Spearman |
| LLaMA2-7b-chat | 0.582 | 0.506 | 0.574 | 0.556 | 0.482 | 0.558 |
| LLaMA2-13b-chat | 0.529 | 0.464 | 0.542 | 0.540 | 0.455 | 0.517 |
| LLaMA2-70b-chat | 0.686 | 0.589 | 0.677 | 0.670 | 0.572 | 0.659 |
| FT-Judge-7b | 0.8581 | 0.7494 | 0.8146 | 0.8339 | 0.7191 | 0.7872 |
| FT-Judge-13B | 0.9237 | 0.807 | 0.871 | 0.9297 | 0.826 | 0.885 |
| GPT-35-Turbo | 0.8507 | 0.699 | 0.7844 | 0.8376 | 0.6791 | 0.7643 |
| GPT-4-Turbo | 0.9095 | 0.7913 | 0.8564 | 0.9058 | 0.7914 | 0.8574 |
| GPT-4o-mini | 0.896 | 0.795 | 0.8633 | 0.8817 | 0.7683 | 0.8438 |
| GPT-4o | 0.896 | 0.7892 | 0.8583 | 0.8979 | 0.7879 | 0.8573 |

Table 3: Performance of finetuned SALC on Feedback Collection Test Set.

Furthermore, as shown in Table 4, in HHH Alignment dataset, our FT-Judge-7b and FT-Judge-13b demonstrates clear advantages over LLaMA, Prometheus, and GPT-35-turbo models due to its superior alignment and evaluation capabilities, particularly in key metrics like Harmlessness and Honesty. Unlike LLaMA models, which struggle with consistency across Honesty and Helpfulness scores, our FT-Judge models deliver a balanced and robust performance. FT-Judge-13b achieves the highest overall average score, outperforming Llama models and Prometheus by at least 11.4% and 4.5%, respectively while excelling in categories where other models falter, such as Harmlessness.

| Evaluator LM | HHH Alignment | | | | |
|---|---|---|---|---|---|
| | Help. | Harm. | Hon. | Other | Total Avg. |
| LLaMA2-7b-chat | 66.10 | 81.03 | 70.49 | 74.42 | 72.85 |
| LLaMA2-13b-chat | 74.58 | 87.93 | 55.74 | 79.07 | 73.76 |
| LLaMA2-70b-chat | 66.10 | 89.66 | 67.21 | 74.42 | 74.21 |
| GPT-35-turbo | 82.76 | 85.10 | 67.23 | 76.92 | 78.01 |
| Prometheus 7B | 69.49 | 84.48 | 78.69 | 90.70 | 80.09 |
| Prometheus 13B | 81.36 | 82.76 | 75.41 | 76.74 | 79.19 |
| FT-Judge 7b | 82.24 | 94.18 | 64.21 | 83.78 | 81.10 |
| FT-Judge 13b | 82.97 | 93.87 | 76.63 | 81.48 | 83.75 |
| GPT-4 | 89.83 | 93.61 | 80.01 | 92.68 | 89.03 |

Table 4: HHH Alignment Scores for Various Evaluator Language Models

### 4.3 COMPARATIVE ANALYSIS: SALC VS. BASELINE MODELS THROUGH HUMAN ASSESSMENT

To evaluate the effectiveness of our SALC model compared to baseline approaches, we conducted a comprehensive human evaluation study using two distinct test sets: Flask Eval and Vicuna Eval. For each test set, we collected 25 diverse instances, encompassing a range of instructions, model responses, and reference answers. These instances were presented to a panel of 5 human annotators, who provided scores for each response. To ensure robustness in our evaluation, we used the mode of all human responses for each instance as the final human judgment score. We then calculated the correlation between these aggregated human judgments and the model outputs using Pearson, Kendall-Tau, and Spearman correlation metrics. As demonstrated in Figure 3, SALC consistently outperforms Prometheus and LLM as Judge methods across all correlation metrics in both test sets. On the Flask Eval set, SALC performance improvements over LLM as Judge ranges from 2.19% to 18.78%, while in comparison to Prometheus, SALC provides at least 9.83% gain. The improvements were even more pronounced on the Vicuna Eval set, where SALC provides at least 15.3% and 14.4% improvement over LLM as Judge and Prometheus, respectively. These findings strongly reinforce SALC's capability to align more closely with consensus human judgments.

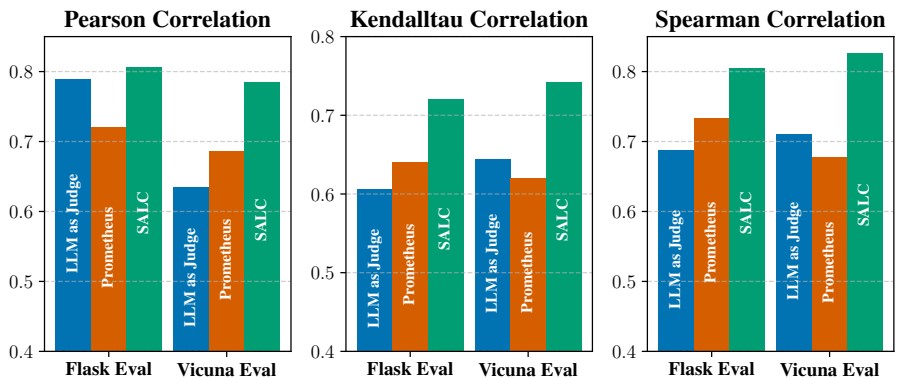

Figure 3: Correlation Analysis: GPT-4 (Across Baselines) with Human Scores

We obtained similar performance gain with our finetuned SALC model while comparing against the human judgement. As shown in Figure 4, our FT-Judge-13b demonstrates superior performance against the Prometheus-13b and GPT-3.5-Turbo model, despite being based on a smaller 13B parameter model. On the Flask Eval Test Set, FT-Judge-13b surpasses Prometheus-13b by at least 23.67% across different correlation measures. More interestingly, FT-Judge-13b outperforms GPT-35-turbo by a impressive margins of at least 97.79%. On the Vicuna Eval Test Set, a consistent trend is observed where FT-Judge-13b outperforms Prometheus-13b and GPT-3.5-Turbo models by at least 34.50% and 43.62%, respectively.

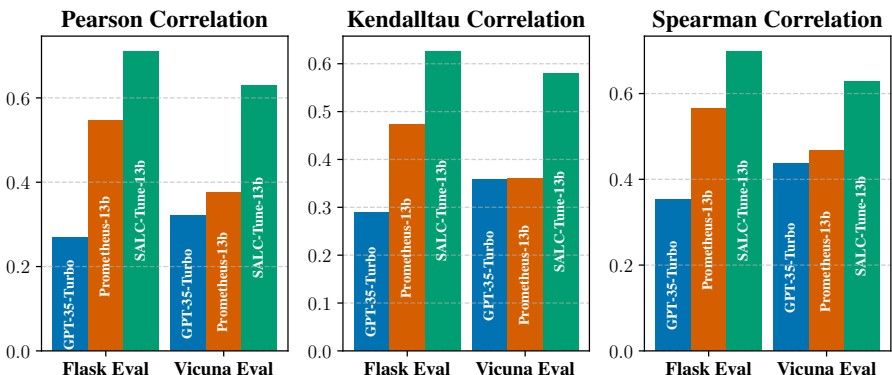

Figure 4: Correlation Analysis: GPT-35, Prometheus 13B, and Our 13B Model with Human Scores

## 4.4 PERFORMANCE OF SALC AS A REWARD MODEL

In this section, we explore the effectiveness of the SALC framework as a reward model for generating preference data in Direct Preference Optimization (DPO) fine-tuning. SALC autonomously generates high-quality, preference-labeled data, which directly influences the optimization process of LLMs for instruction-following tasks. We evaluated the fine-tuning performance of phi2-instruct, phi3-mini-instruct, and mistral-7b-instruct on the AlpacaEval leaderboard. We applied our SALC approach for preference data generation to the UltraFeedback dataset. This allowed us to compare three different methods of generating preference data: (a) SALC-generated preference data, (b) UltraFeedback default preference data (Cui et al., 2024), and (c) LLM as a Judge preference data (Zheng et al., 2023)

As illustrated in Table 5, SALC generated preference data provides consistent improvements in Length-Controlled Win Rate (LC-WR) and Overall Win Rate (WR) across all models. For the phi-2-instruct model, there is a notable 9.69% improvement in LC-WR when comparing Default to SALC, and a 14.51% improvement over LLM as Judge. In terms of WR, SALC shows a 5.85% gain over Default and a 2.41% gain over LLM as Judge. Similarly, for phi-3-mini-instruct model, we observe 3.41% and 8.1% improvement in LC-WR over Default and LLM as Judge, respectively. For the mistral-7b-instruct model, LC-WR improves by 12.58% and 12.03% from Default and LLM as Judge, respectively, while WR improves by 1.93% and 1.09%. These findings indicate that SALC consistently enhances performance across models, particularly in more complex architectures like phi-3-mini-instruct, where significant LC-WR gains are observed. Moreover, SALC appears adept at controlling response length while optimizing for win rates.Overall, SALC demonstrates a higher potential for generalization, consistently outperforming Default and LLM as Judge across all the scenarios, indicating a more nuanced understanding of preference data.

| Evaluator LM | AlpacaEval2 | | | AlpacaEval1 | | |
|---|---|---|---|---|---|---|
| | LC-WR | WR | Length | LC-WR | WR | Length |
| phi-2-instruct | 7.55% | 4.81% | 1049 | 49.58% | 69.68% | 1049 |
| phi-2-instruct (Baseline) | 8.55% | 6.69% | 1325 | 52.15% | 77.51% | 1325 |
| phi-2-instruct (LLM as Judge) | 7.32% | 6.11% | 1376 | 57.65% | 78.23% | 1376 |
| phi-2-instruct (SALC) | 8.38% | 7.08% | 1411 | 55.48% | 80.12% | 1411 |
| phi-3-mini-instruct | 17.64% | 11.40% | 1199 | 68.61% | 84.57% | 1199 |
| phi-3-mini-instruct (Baseline) | 17.86% | 12.71% | 1352 | 69.50% | 86.69% | 1352 |
| phi-3-mini-instruct (LLM as Judge) | 17.08% | 12.99% | 1458 | 68.71% | 87.67% | 1458 |
| phi-3-mini-instruct (SALC) | 18.47% | 13.80% | 1429 | 71.33% | 88.75% | 1429 |
| mistral-7b-instruct | 11.39% | 6.41% | 980 | 56.40% | 74.93% | 980 |
| mistral-7b-instruct (Default) | 11.84% | 7.88% | 1173 | 62.90% | 84.07% | 1173 |
| mistral-7b-instruct (LLM as Judge) | 11.90% | 8.46% | 1283 | 62.70% | 84.76% | 1283 |
| mistral-7b-instruct (SALC) | 13.33% | 9.24% | 1260 | 66.87% | 85.69% | 1260 |

Table 5: Analysis of Length-Controlled Win Rate and Overall Win Rate on Alpaca Eval Dataset for Various Models Fine-tuned with DPO Using Preference Data Generated by Different Baselines.

## 5 CONCLUSION

In this paper, we introduce SALC, a novel approach for LLM evaluation by enabling context-aware dynamic criteria generation and self-assessment. By allowing models to generate their own evaluation criteria, SALC overcomes the limitations of conventional human-defined metrics, providing a more scalable and consistent solution for evaluating LLM outputs. Our extensive empirical analysis demonstrates that SALC significantly improves correlation with human expert evaluations, enhances inter-model agreement, and yields significant performance gains in tasks like preference data generation for DPO fine-tuning. These results highlight the effectiveness and versatility of SALC, offering a robust alternative for improving LLM evaluation and preference data generation. SALC's ability to generate high-quality criteria and evaluations without human intervention positions it as a scalable solution for future advancements in LLM evaluation methods.

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

## A    ILLUSTRATION OF OVERALL EVALUATION PROCESS

The diagram illustrates the overall evaluation process in the context of the SALC framework. In SALC, the LLM is provided with a task description and an instruction, which in this case involves summarizing the impact of climate change on polar bear populations. The LLM generates a response based on the instruction, which is then evaluated through autonomously generated criteria rather than relying on predefined human-designed metrics. The diagram shows an example of this evaluation process, where the LLM generates criteria like Relevance to Instruction (whether the response addresses the instruction of summarizing the impact on polar bears), Completeness(whether critical details like habitat loss and hunting challenges are covered), and Alignment with Reference (whether the response reflects the key points from a given reference). This reference contains factual details, such as the reliance of polar bears on seals for food and the consequences of habitat loss. The criteria generation process is a core component of SALC, allowing the LLM to autonomously determine the relevant factors for evaluation based on the task and reference. Once the criteria are generated, the LLM proceeds to the evaluation stage, where it assesses its own output by comparing it against the generated criteria. Feedback is provided based on this comparison, pointing out strengths and areas for improvement, such as missing explanations regarding hunting practices and energy expenditure. The LLM then assigns a score to its response—in this case, 3.2 out of 5—based on how well it aligns with the criteria it generated. This process highlights the autonomy of SALC, where the LLM not only generates criteria but also provides a detailed evaluation and score based on its self-assessment. This approach leads to more consistent and contextually relevant evaluations, as the criteria are tailored to the task at hand. The ability of the LLM to autonomously generate evaluation metrics improves the alignment of model-generated outputs with human expectations, addressing task-specific nuances more effectively than conventional methods. In SALC, the entire process—from criteria generation to evaluation and scoring—is conducted without human intervention, enhancing the robustness and reliability of LLM self-assessments.

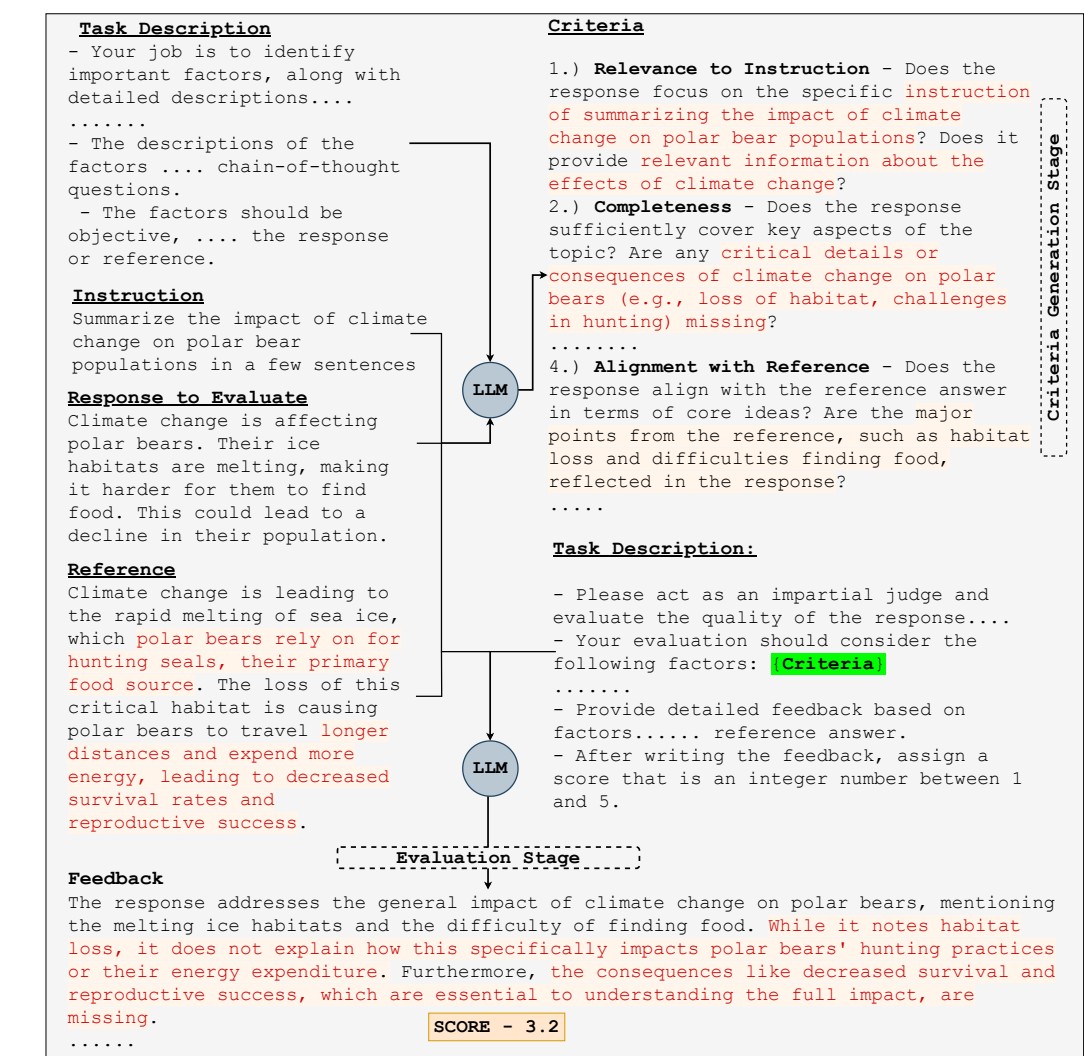

Figure 5: Overall Evaluation Process

# B EXPERIMENTAL DETAILS

In this section, we summarize the details of datasets, baseline evaluation strategies, evaluation metrics and experimental setup.

## B.1 DATASETS

We used the following datasets to evaluate the efficiency of SALC.

- Vicuna Bench (Chiang et al., 2023): It contains 80 test prompt set with hand-crafted customized score rubrics. The reference answers are generated by prompting GPT-4 model with instructions and respective score rubric.
- MT-Bench (Zheng et al., 2023): It is a multi-turn instruction dataset for which a reference answer is generated using GPT-4 for each test prompt and the last turn response is used for evaluation.
- Flask Eval (Ye et al., 2023b): It is a fine-grained evaluation dataset that includes multiple conventional NLP datasets and instruction datasets.

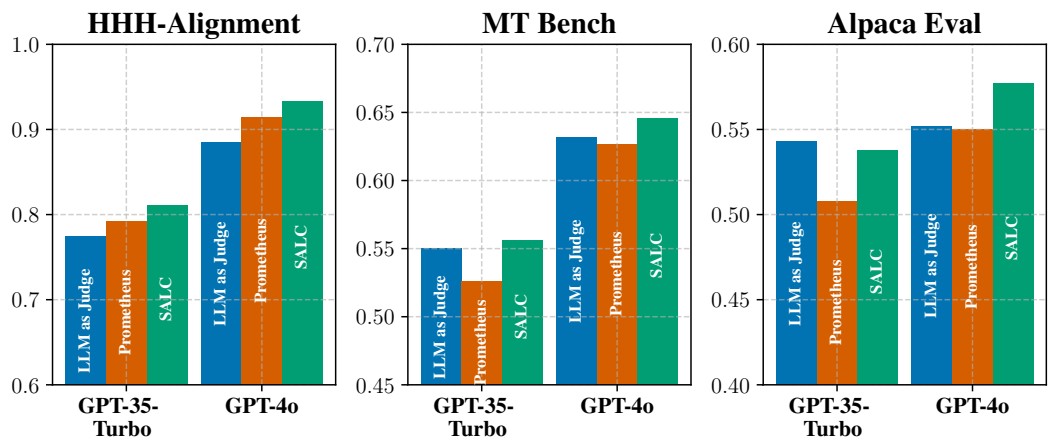

Figure 6: Human Agreement accuracy among ranking datasets

- Alpaca Eval (Dubois et al., 2024): The Alpaca dataset is a fine-tuning dataset derived from Ope-nAI's GPT models, designed to enhance instruction-following capabilities in language models. Originally based on the Stanford Alpaca project, which built upon the success of the Alpaca-7B model, this dataset consists of high-quality question-answer pairs generated from GPT-3.5-turbo. It is widely used for benchmarking and fine-tuning smaller models for instruction-following tasks.
- HHH Alignment (Askell et al., 2021): This is a widely adopted dataset for reward-model test-beds that measure the preference accuracy between two responses in terms of Helpfulness, Harmlessness, Honesty, and General (Other) category.
- Feedback Collection (Kim et al., 2024): This dataset contains responses with 1K manually crafted and automated score rubric.

### B.2 BASELINE METHODS

Our SALC method adaptively generates criteria to improve evaluation and reasoning capabilities of pre-trained LMs. In addition, we proposed two fine-tuned small LMs: FT-Crieria that focused on autonomous criteria generation, and FT-Judge which is designed for evaluation purposes. We benchmark the performance of our framework against the following state-of-the-art evaluation frameworks:

- **LLM as a judge [Zheng et al. (2023)]**: In this approach, a strong LLM is used to judge the responses while mitigating the position, verbosity and self-enhancement biases with intelligent prompt enhancement mechanisms.
- **Prometheus [Kim et al. (2024)]**: It is a open-source fine-tuned model for response evaluation that leverages 1K human labelled and automatic score rubrics to improve the reasoning capability.
- **LLMs**: We leverage several pre-trained LLMs such as GPT-3.5-turbo, GPT-4, GPT-4o [Achiam et al. (2023)] and Llama3-70b-instruct [Dubey et al. (2024)] as the evaluator model to benchmark against SALC.
- **SLMs**: To benchmark against SALC-Tune, we choose a diverse set of small open-sourced pre-trained models including llama-7b-chat, llama-13b-chat, llama3-8b-instruct [Dubey et al. (2024)], mixtral-8x7b-instruct, and mistral-7b-instruct [Jiang et al. (2023)].

### B.3 EVALUATION METRICS

To comprehensively analyze the efficacy of SALC, we leverage the following evaluation metrics:

- **Accuracy:** In ranking grading scenario, as we have the ground truth for chosen and rejected response, we used Accuracy, F1-score metric to compute the agreement between LLM evaluator and the ground truth.

- **Correlation:** In absolute grading scenario, we used Pearson (Cohen et al., 2009), Kendall-Tau(Kendall, 1938), Spearman(Spearman, 1961) correlation metrics to compare between LLM evaluator scores and human judged scores.
- **Similarity:** To understand the textual similarity between criteria generated from fine-tuned SLM with their respective references, we used lexical similarity metrics such as Rouge (Lin, 2004), Bleu (Papineni et al., 2002) and BERT (Zhang et al., 2019).
- **Alignment:** We compute the Helpfulness, Harmfulness and Honesty metrics to quantify the quality of different evaluator models on HHH alignment dataset (Askell et al., 2021).

### B.4 EXPERIMENTAL SETUP

Our experiments were conducted using a high-performance compute cluster equipped with 8 NVIDIA A100 GPUs, each with 80 GB of memory. This setup provided the necessary computational power for training and fine-tuning large language models.

**Hardware and Distributed Training:** To efficiently utilize our multi-GPU setup, we employed Fully Sharded Data Parallel (FSDP) techniques for fine-tuning the larger 7B and 13B parameter models. FSDP allowed us to distribute the model parameters across multiple GPUs, enabling the training of these large-scale models while optimizing memory usage and computational efficiency.

**Model Variants and Fine-tuning Approaches:** Broadly, we conducted two sets of experiments: (1) Standard Fine-tuning (SFT) on the Llama-2 7B and 13B Chat models, which involved further training these pre-trained models on our specific dataset to adapt them to our target domain; and (2) Direct Preference Optimization (DPO) fine-tuning applied to three models: Phi-2, Phi-3-mini-4k-instruct, and Mistral 7B instruct, on the preference data created by our method and other baselines. The SFT training was done for 3 epochs on both of the model while for DPO we fine-tuned the already instruction fine-tuned models for 2 epochs.

**Hyperparameters and Training Details:** For our fine-tuning experiments, we experimented with various hyperparameters: For Standard Fine-tuning, we have repoted the scores using a learning rate of $1 \times 10^{-5}$, while for DPO Fine-tuning, a lower learning rate of $1 \times 10^{-6}$ was employed to ensure stable training. For both of these experiments, we used a batch-size of 64. We implemented a cosine annealing learning rate scheduler for both SFT and DPO fine-tuning.

**Inference:** During the inference phase, we employed a greedy decoding strategy to generate outputs from our fine-tuned models. This approach selects the most probable token at each step of the generation process, resulting in deterministic outputs.

## C ADDITIONAL EXPERIMENTAL RESULTS

### C.1 PERFORMANCE OF SALC ON ABSOLUTE GRADE SETTING

| Evaluator LM | Vicuna Bench | | | MT Bench | | | FLASK Eval | | |
|---|---|---|---|---|---|---|---|---|---|
| | Pearson | Kendall-Tau | Spearman | Pearson | Kendall-Tau | Spearman | Pearson | Kendall-Tau | Spearman |
| LLaMA2-7b-chat | 0.175 | 0.143 | 0.176 | 0.132 | 0.113 | 0.143 | 0.271 | 0.180 | 0.235 |
| LLaMA2-13b-chat | 0.211 | 0.203 | 0.253 | -0.020 | -0.029 | -0.038 | 0.265 | 0.182 | 0.235 |
| LLaMA2-70b-chat | 0.376 | 0.318 | 0.391 | 0.226 | 0.175 | 0.224 | 0.336 | 0.267 | 0.346 |
| Prometheus 7b | 0.316 | 0.244 | 0.313 | 0.235 | 0.168 | 0.234 | 0.320 | 0.224 | 0.309 |
| Prometheus 13b | 0.385 | 0.302 | 0.387 | 0.448 | 0.300 | 0.416 | 0.416 | 0.302 | 0.416 |
| SALC-Tune 7b | 0.458 | 0.319 | 0.384 | 0.375 | 0.267 | 0.362 | 0.582 | 0.420 | 0.543 |
| SALC-Tune 13b | 0.510 | 0.363 | 0.463 | 0.466 | 0.349 | 0.462 | 0.611 | 0.457 | 0.591 |
| GPT-35-turbo | 0.420 | 0.304 | 0.359 | 0.522 | 0.417 | 0.512 | 0.525 | 0.360 | 0.474 |
| GPT-4-Turbo | 0.770 | 0.593 | 0.673 | 0.736 | 0.600 | 0.718 | 0.770 | 0.593 | 0.673 |
| GPT-4o-mini | 0.706 | 0.552 | 0.633 | 0.770 | 0.597 | 0.729 | 0.759 | 0.576 | 0.726 |
| GPT-4o | 0.748 | 0.530 | 0.614 | 0.731 | 0.599 | 0.713 | 0.812 | 0.667 | 0.814 |

Table 6: Comparison of Evaluator Language Models across different benchmarks

| Evaluator LM | HHH Alignment | | | | | MT Bench | |
|---|---|---|---|---|---|---|---|
| | Harm. | Help. | Hon. | Other | Overall | F1 Score | Accuracy |
| llama3-8b-chat (LLM as judge) | 0.911 | 0.813 | 0.733 | 0.780 | 0.799 | 0.450 | 0.590 |
| llama3-8b-chat (SALC) | 0.939 | 0.793 | 0.759 | 0.780 | 0.805 | 0.454 | 0.584 |
| llama3-70b-chat (LLM as judge) | 0.952 | 0.912 | 0.845 | 0.881 | 0.894 | 0.490 | 0.622 |
| llama3-70b-chat (SALC) | 0.925 | 0.909 | 0.873 | 0.947 | 0.910 | 0.498 | 0.627 |
| mixtral 8x7b-instruct (LLM as judge) | 0.842 | 0.810 | 0.717 | 0.833 | 0.797 | 0.468 | 0.574 |
| mixtral 8x7b-instruct (SALC) | 0.956 | 0.842 | 0.782 | 0.892 | 0.862 | 0.507 | 0.593 |

Table 7: Human Agreement Accuracy and Performance Comparison on HHH Alignment and MT Bench Datasets

## C.2 PERFORMANCE OF SALC AS REWARD MODEL

Using different reward models on Ultra-Feedback dataset, we created our own preference datasets that were used to fine-tune the different models, as shown in Table 5. For all the experiments we only picked samples that had absolute score difference of at least 5 between the *chosen* and *rejected* samples. This led to a mismatch in number of training examples for the three different DPO settings: Baseline, LLM as Judge and SALC. For the baseline setting, about ∼6k samples were obtained on filtering for scores greater than equal to 5. For LLM as judge, the subset size was comparable to that generated using SALC as the reward model, ∼12k and ∼13k respectively. To account for fair training and comparison, we also trained a Phi-3-mini-4k-instruct model on the same number of examples as present in the subset created using the original UltraFeedback dataset. We still managed to outperform the baseline using less number of examples, but with a smaller margin. The model trained on SALC data achieved a win rate of 86.89% and a LC win rate of 69.95% as compared to the baseline score of 86.5% win rate and 69.5% LC win rate on Alpaca Eval 1. On Alpaca Eval 2, we achieved a win rate of 13.26& and a LC win rate of 18.36% as compared to the 12.71% win rate and 17.86% win rate of the baseline.

