# OpenReview forum: "Unveiling Context-Aware Criteria in Self-Assessing LLMs"
_ICLR.cc/2025/Conference — ICLR 2025 Conference Withdrawn Submission_

### Official Review · Reviewer_TD9Q · 2024-10-17

**Soundness:** 1
**Presentation:** 2
**Contribution:** 1
**Rating:** 3
**Confidence:** 4

**Summary:**

This paper introduces SALC, a framework that leverages large language models (LLMs) to evaluate model responses, either in an absolute or relative manner. Unlike previous work, SALC first generates sample-specific evaluation criteria, which aim to accurately capture the most relevant aspects of the input. After these criteria are created, the judge LLM is asked to either provide an absolute score for a model response on a scale of 1 to 5 (absolute setting) or to choose the better response between two options (relative setting). The paper demonstrates that SALC outperforms the included baselines across a variety of benchmarks and also shows that it can generate preference data for DPO leading to higher evaluation scores on AlpacaEval compared to other methods.

**Strengths:**

- The authors perform evaluations across various benchmarks and settings, demonstrating that their method outperforms the provided baselines.
- Despite minor grammatical mistakes and typos, the paper is clear and easy to follow.
- The approach of generating sample-specific evaluation criteria has not been previously applied in this context and could improve automatic evaluation of model responses.

**Weaknesses:**

While the paper proposes a new prompting strategy, there is no substantial technical novelty. A lack of technical novelty alone does not justify rejection, but it does require a paper to be impeccable regarding its experimental results and explanations. However, the paper suffers from several major weaknesses in their experimental results:
- No code is included in the supplementary material, making it difficult to verify the reproducibility of the experiments. Additionally, some experimental details are missing in the paper (e.g., prompts used for SALC and the baselines), which could be resolved by providing the code and providing more experimental details in the paper.
- Confidence intervals, such as those computed through bootstrapping, are not reported in any of the experiments. This is particularly concerning in Section 4.3, where only 25 samples are used for each correlation measurement.
- Appendix C.2 highlights an important detail that is never mentioned in the main text. The paper does not explain why there is such a large difference in the number of training samples for each method. It would be more appropriate to remove the score difference requirement so that all models are trained on the same dataset. Since their method now trains on more data than the baselines, this is a significant issue. Appendix C.2 seemingly performs such an experiment, but the results are much less convincing and not discussed in full detail.
- Some baselines are missing from certain parts of the paper. For example:
     - Lines 317–319 state that the authors’ models outperform larger state-of-the-art open-source models, but no such comparison is actually presented.
     - Prometheus is missing from Table 3, and the paper does not specify whether Prometheus or Prometheus 2 was used, making the comparison unclear. This should be clarified.
- Tables 2 and 3 use inappropriate metrics. As indicated in Table 1, GPT-4o outperforms GPT-4. However, Tables 2 and 3 suggest that FT-Judge (a 13B Llama-2 model trained on GPT-4 outputs) performs better than GPT-4o, which is highly unlikely. The experiment in Table 1 should be repeated for both the Prometheus and FT-Judge models to show correlation with human judgment instead of correlation with GPT-4.
- Llama-2 models are outdated. The experiments in Table 1, 2, 3, and 4 should be performed on more recent models (e.g., Llama-3.1/3.2).
- RewardBench [1] is a recently introduced comprehensive benchmark for evaluating reward models. However, no comparison is made between SALC(-Tune) and state-of-the-art methods for this benchmark. Since these methods are much more varied than the baselines in the paper, this further decreases confidence that the paper is actually outperforming state-of-the-art. Thus, the authors should evaluate their finetuned models (and SALC) on RewardBench and check they outperform models of the same size and architecture that are publicly available on https://huggingface.co/spaces/allenai/reward-bench.

Furthermore, the rest of the paper also contains some issues:
- The paper does not mention WildBench [2], a well-known benchmark that also incorporates sample-specific evaluation criteria. While there are some differences between SALC and WildBench, WildBench is clearly related to SALC and therefore must be discussed.
- Section 4.3 includes a human evaluation experiment, but details about the process are missing. This violates the ICLR Code of Ethics, which states that research involving human subjects must include the approval of an ethical review board. The paper should mention any relevant ethical approvals and provide a detailed description of the evaluation process.

Other minor comments:
- Section 4.4 reports relative gains, while Table 5 shows absolute numbers. This gives the impression that the results are better than they actually are. The authors should clearly state that relative gains are discussed in the paper, and explain their reason for doing so instead of reporting absolute gains (as is more common). Since these results are also reported in the abstract and introduction, this can give a wrong impression to the reader.
- The paper contains several typos that should be addressed before publication (e.g., Lines 38–39, 231–234, 263–267, 309, 323, 340, 506), along with the use of an undefined symbol on Line 302. Employing a grammar checking tool could significantly enhance the overall presentation and clarity.

[1] Lambert, Nathan, et al. "Rewardbench: Evaluating reward models for language modeling." arXiv preprint arXiv:2403.13787 (2024).

[2] Lin, Bill Yuchen, et al. "WILDBENCH: Benchmarking LLMs with Challenging Tasks from Real Users in the Wild." arXiv preprint arXiv:2406.04770 (2024).

**Questions:**

- How is $\beta_j$ computed “internally”? This point is rather vague and needs to be clarified.
- Models are instructed to give a score between 1 and 5. In Appendix C.2, you mention that the absolute score difference must be at least 5. How is this possible if the range of scores is just 4 ($=5-1$)?
- Based on the description, it seems that the generated evaluation criteria might be not only input-specific but also output-specific. This raises the possibility that models are being evaluated based on different criteria for the same question. If this is correct, have the authors checked whether this is the case? Additionally, if different criteria are used for the same question, have the authors investigated whether this significantly impacts the results?

**Details Of Ethics Concerns:**

The authors present a small human study in Section 4.3. The experiment itself is rather innocuous, but no further details about the experimental setup is provided. Furthermore, the ICLR Code of Ethics states that an approval by an ethics review board must be obtained for human experiments, and none is provided in the paper. Since the experiment is so small, I am not sure whether this constitutes unresponsible research practice. Therefore, letting an expert look into this problem seems appropriate.

---

### Official Review · Reviewer_K3Cz · 2024-11-01

**Soundness:** 3
**Presentation:** 3
**Contribution:** 2
**Rating:** 5
**Confidence:** 4

**Summary:**

This paper presents SALC, a new evaluation framework that enables LLMs to first generate instance-level evaluation criteria based on context and then conduct assessments accordingly. By supporting both absolute scoring and relative preference evaluation settings, SALC provides a comprehensive solution for model evaluation. Experimental results demonstrate SALC significantly outperformers existing methods. Moreover, the authors fine-tune small models to distill criteria generation and assessment abilities from GPT4.Beyond evaluation tasks, SALC also proves effective in generating high-quality preference data.

**Strengths:**

1.	The paper introduces a new approach to LLM evaluation by enabling dynamic, context-aware criteria generation, which fundamentally differs from traditional static evaluation methods.
2.	The comprehensive experiments across multiple benchmarks demonstrate consistent and significant improvements over strong baselines.
3.	SALC can be used to generate high-quality preference data, making it a versatile tool for model development and fine-tuning.

**Weaknesses:**

1.	The paper lacks a systematic analysis of the model-generated criteria. A more thorough examination of how these dynamically generated criteria differ from static ones and why they lead to better evaluation outcomes would strengthen the paper's claims about SALC's advantages.
2.	The paper lacks sufficient detail about how multiple criteria are weighted in the evaluation process. Does the model generate these weights simultaneously with the criteria generation process?

**Questions:**

1.	Tables 2 and 3 in Section 4.2 show that FT-Criteria's outputs are highly correlated with GPT-4, with correlation levels even exceeding those of GPT-4-turbo and GPT-4o. However, this correlation measurement with GPT-4 is not particularly meaningful since FT-Criteria was trained through knowledge distillation from GPT-4. Furthermore, since GPT-4-turbo and GPT-4o can be considered more powerful models than GPT-4, FT-Criteria's higher correlation with GPT-4 compared to these two models does not demonstrate any significant advantage.
2.	While section C.4 addresses the data imbalance issue in baseline comparisons and provides results from "Phi-3-mini-4k-instruct" trained with matched sample sizes, the evaluation methodology could be more rigorous. A more standard approach would be to compare the models after training on the complete UltraFeedback dataset, rather than on filtered subsets. This would provide a more comprehensive and fair assessment of SALC's effectiveness as a reward model.
3.	In the relative setting, the LLM can only see both responses during the criteria generation phase, not during the assessment phase. Does this design choice impact the final evaluation quality? Would the performance improve if the model could compare both responses simultaneously during the scoring phase?

---

### Official Review · Reviewer_Sf5r · 2024-11-04

**Soundness:** 2
**Presentation:** 2
**Contribution:** 2
**Rating:** 3
**Confidence:** 4

**Summary:**

The paper proposes Self-Assessing LLM with Autonomous Criteria Generation (SALC) that enhances the evaluation capabilities of large language models (LLMs) in assessing generated text. The core idea behind SALC is to enable LLMs to autonomously generate context-aware evaluation criteria tailored to specific instances rather than relying on static, human-defined metrics. This innovative approach leverages dynamic knowledge to improve the evaluation of generative tasks across diverse datasets. Empirical results show that SALC significantly outperforms existing evaluation frameworks, with improvements demonstrating its effectiveness as a scalable and robust evaluation method in NLP.

**Strengths:**

1. SALC's ability to generate dynamic, context-specific evaluation criteria is a significant advancement over traditional evaluation methods. This approach allows for more nuanced and task-appropriate assessments of LLM outputs.
2. The authors demonstrate that smaller, fine-tuned models can perform comparably to much larger models like GPT-4 in generating evaluation criteria and feedback. This efficiency is crucial for practical applications and wider adoption.
3. The evaluation of SALC using multiple datasets (Vicuna Bench, MT-Bench, and Flask Eval) provides a robust assessment of the framework's performance across different types of tasks and data structures.

**Weaknesses:**

1. A significant weakness of the SALC framework is the lack of evaluation of the LLM-generated criteria themselves. While the authors use LLM-generated criteria as a key feature of their method, they do not adequately assess the effectiveness and fairness of these generated criteria. This is a crucial oversight, as the quality of the evaluation heavily depends on the quality of these criteria. Although Table 2 in the paper provides some analysis using metrics like BERTScore to compare generated criteria with a fixed set, this comparison is insufficient to fully validate the fairness and effectiveness of the generated criteria. The evaluation of these criteria should be as thorough and important as the evaluation of the final answers themselves.
2. The paper lacks detailed information about hyperparameters and the availability of code or data. Providing more specific details about the experimental setup and making the code publicly available (if not already done) would enhance reproducibility.
3. The paper could address potential biases that might be introduced in the criteria generation process. A discussion on how the framework mitigates or accounts for potential biases inherent in the base LLM would be valuable.
4. There are a lot of typos in the paper.

**Questions:**

Are there plans to release the code and fine-tuned models used in this study to facilitate reproducibility and further research in this area?
How does SALC handle potential inconsistencies or contradictions in generated criteria across different evaluation instances for similar tasks?

---

### Official Review · Reviewer_4fLt · 2024-11-04

**Soundness:** 1
**Presentation:** 2
**Contribution:** 1
**Rating:** 3
**Confidence:** 4

**Summary:**

This paper proposed an approach SALC for LLM evaluation by conditioning on instance-level criteria, which is generated by the LLM itself. The paper also SFT a small LM with GPT-4 distilled criteria and judgements and showed the sft-ed models outperform the zero-shot larger ones.
The paper also claims SALC can be used to generate preference data for DPO, but the details is not explained.

**Strengths:**

- The writing of this paper is acceptable.
- The experiment results shows good improvements, which means the method might work.

**Weaknesses:**

- The writing: Although the presentation of the paper looks good, but there are a few places quite confusing.
    - Why is there no a specific section about the actual method used for generating preference data? Only mentioned in Experiment 4.4, but not the actual data generation approach.
    - The term of SALC is over-used. It seems SALC was referred as multiple things, such as an evaluation method, the preference data generation method, the trained model, or even the whole framework?
    - Some paragraphs should be structured better for the purpose of reading. e.g. Line 337-357. Lin 435-447.
    - Line 302, what's L? Is it the teacher model GPT-4? Did you mentioned before?
    - Line 327, what exactly is your LLM-as-judge approach? Direct scoring? CoT scoring? G-Eval? Pairwise comparison?

- Novelty: In general, I think this paper has very limited novelty.
    - Method Section 3.1 Criteria Generation: Generating criteria based on the instruction & response/reference has very largely been studied and used in Automatic Prompt Optimization area (textual feedbacks)
    - Method Section 3.2 Absolute assess and Relative assess are just the basic pointwise and pairwise evaluations, which has been widely used in the domain of LLM-as-judge.
    - Method Section 3.3 SFT a smaller LMs with GPT-4's feedback and judgements. This is just standard data distillation. A distill-sft small LLM outperform zero-shot LLMs in a specific dataset is not surprising results.

- Motivation: In general, I feel the motivation is strange.
    - You need reference to generate criteria for each instance. Then you show that with criteria the evaluation is more aligned with human compared to reference-free approaches. This is not a fair comparison.
    - If you rely on different criteria for each instance, when the evaluation tasks is a bit subjective, such as evaluating summaries, the generated criteria will show preference to the specific style of the provided reference. The evaluation is then not fair anymore.

**Questions:**

- In your Table 1, does the SALC evaluation method see the reference before generating criteria?
- In Table 5, Is your SALC model fine-tuned? Or just provided with criteria? If yes to any of them, why this is a fair comparison with the zero-shot models?

---

### Note · Authors · 2024-11-20

I have read and agree with the venue's withdrawal policy on behalf of myself and my co-authors.